# Laminarin Reduces Cholesterol Uptake and NPC1L1 Protein Expression in High-Fat Diet (HFD)-Fed Mice

**DOI:** 10.3390/md21120624

**Published:** 2023-11-29

**Authors:** Zhuoqian He, Zhongyin Zhang, Pengfei Xu, Verena M. Dirsch, Limei Wang, Kewei Wang

**Affiliations:** 1Department of Pharmacology, School of Pharmacy, Qingdao University Medical College, Qingdao 266073, China; 2020021146@qdu.edu.cn (Z.H.); xupengfeiqdu@163.com (P.X.); wangkw@qdu.edu.cn (K.W.); 2Department of Medicinal Chemistry, School of Pharmacy, Qingdao University Medical College, Qingdao 266073, China; zyzhang@qdu.edu.cn; 3Department of Pharmaceutical Sciences, Faculty of Life Sciences, University of Vienna, 1090 Vienna, Austria; verena.dirsch@univie.ac.at; 4Institute of Innovative Drugs, Qingdao University, Qingdao 266071, China

**Keywords:** laminarin, high-fat diet, cholesterol uptake, NPC1L1

## Abstract

Aberrantly high dietary cholesterol intake and intestinal cholesterol uptake lead to dyslipidemia, one of the risk factors for cardiovascular diseases (CVDs). Based on previous studies, laminarin, a polysaccharide found in brown algae, has hypolipidemic activity, but its underlying mechanism has not been elucidated. In this study, we investigated the effect of laminarin on intestinal cholesterol uptake in vitro, as well as the lipid and morphological parameters in an in vivo model of high-fat diet (HFD)-fed mice, and addressed the question of whether Niemann–Pick C1-like 1 protein (NPC1L1), a key transporter mediating dietary cholesterol uptake, is involved in the mechanistic action of laminarin. In in vitro studies, BODIPY-cholesterol-labeled Caco-2 cells were examined using confocal microscopy and a fluorescence reader. The results demonstrated that laminarin inhibited cholesterol uptake into Caco-2 cells in a concentration-dependent manner (EC_50_ = 20.69 μM). In HFD-fed C57BL/6J mice, laminarin significantly reduced the serum levels of total cholesterol (TC), total triglycerides (TG), and low-density lipoprotein cholesterol (LDL-C). It also decreased hepatic levels of TC, TG, and total bile acids (TBA) while promoting the excretion of fecal cholesterol. Furthermore, laminarin significantly reduced local villous damage in the jejunum of HFD mice. Mechanistic studies revealed that laminarin significantly downregulated NPC1L1 protein expression in the jejunum of HFD-fed mice. The siRNA-mediated knockdown of NPC1L1 attenuated the laminarin-mediated inhibition of cholesterol uptake in Caco-2 cells. This study suggests that laminarin significantly improves dyslipidemia in HFD-fed mice, likely by reducing cholesterol uptake through a mechanism that involves the downregulation of NPC1L1 expression.

## 1. Introduction

Cardiovascular diseases (CVDs) are the leading cause of death worldwide [1]. Several hypotheses have been implicated in the development and progression of CVDs, including lipid infiltration, thrombosis, damage response, oxidative stress, and inflammation. Among these, the lipid infiltration hypothesis is currently the most accepted, and this hypothesis argues that dyslipidemia plays a significant role in atherogenesis and progression [2]. In response to this hypothesis, the first statin was developed to lower plasma lipid levels, marking a new era in the prevention and treatment of CVDs. Statins competitively inhibit the rate-limiting enzyme of cholesterol synthesis, the 3-hydroxy-3-methyl-glutaryl-CoA reductase (HMGCR), in vivo, thereby reducing endogenous cholesterol synthesis [3]. As a result, the liver compensatorily increases the uptake of low-density lipoprotein cholesterol (LDL-C), effectively clearing LDL-C from the blood. However, statins can lead to severe adverse events, and some patients may experience various degrees of muscle damage, such as myositis, myonecrosis, rhabdomyolysis, and occasionally diabetes and possible hemorrhagic stroke [4,5]. Therefore, it is of great significance to develop novel hypolipidemic drugs.

The concept of Medicine Food Homology (MFH) refers to foods that exhibit medicinal properties and can also act as drugs [6,7]. Novel natural products from MFH with lipid-lowering activity have garnered significant attention, and their constituents can be classified into five main categories: polysaccharides, flavonoids, steroidal saponins, quinones, and alkaloids [8]. Laminarin (Figure 1), also known as kelch starch or brown algal starch, is a polysaccharide found in brown algae [9]. Extensive literature has established that laminarin possesses biological activities, including immune modulation, blood glucose, and blood pressure reduction, as well as anti-tumor, anti-oxidant, anti-inflammatory, and skin barrier repair properties [10,11,12,13]. Furthermore, studies have shown that laminarin can reverse metabolic disorders such as dyslipidemia and atherosclerosis in mice induced by a high-fat diet (HFD) [14,15,16]. Some reports also suggest that laminarin improves the lipid profile and ameliorates atherosclerosis in vivo through decreasing levels of total triglycerides (TG), total cholesterol (TC), LDL-C, while increasing levels of high-density lipoprotein cholesterol (HDL-C) [17]. However, the specific mechanism of action of laminarin is still inconclusive.

Niemann–Pick C1-like 1 protein (NPC1L1) is a crucial transporter for cholesterol uptake and absorption at the villus brush border of the small intestine [18,19]. Its primary function is to facilitate the uptake of cholesterol from the small intestinal lumen into epithelial cells. Additionally, NPC1L1 promotes hepatic cholesterol storage and elevates lipid levels by facilitating cholesterol transport through the small intestine to the liver [20,21]. Ezetimibe is currently the first and only marketed inhibitor of NPC1L1 [22]. By blocking this transporter, ezetimibe effectively inhibits cholesterol uptake in the intestine, reduces hepatic cholesterol storage, and, consequently, lowers plasma cholesterol levels [23]. It has been reported that extracts from brown algae, particularly polysaccharides, have the ability to increase cholesterol excretion in feces [17]. However, whether the cholesterol-lowering effect of brown algae polysaccharides is associated with NPC1L1 has not yet been clarified. Therefore, the purpose of this study is twofold: (i) to investigate whether laminarin, like ezetimibe, can inhibit cholesterol uptake and lower blood lipids by targeting NPC1L1; and (ii) to examine the mechanism underlying the blood lipid-lowering effect of laminarin by utilizing a HFD model of mice.

## 2. Results

### 2.1. Laminarin Reduces Cholesterol Uptake into Caco-2 Cells

First, we assessed the cytotoxicity of laminarin on Caco-2 cells by treating the cells with different concentrations of laminarin (3–100 µM) for 24 h. Cell viability was determined using the resazurin conversion assay. As shown in Figure 2A, laminarin had no significant inhibitory effect on Caco-2 cell viability.

Next, we employed confocal imaging to visualize the effect of laminarin on cholesterol uptake in Caco-2 cells utilizing the fluorescence of BODIPY-cholesterol. The results showed that treatment with 50 μM and 100 μM laminarin significantly reduced the mean fluorescence intensity in Caco-2 cells, as compared to the control. Moreover, the effect of 50 µM laminarin was comparable to the positive control of ezetimibe (Eze, 50 µM) (Figure 2B,C). To further quantify the impact of laminarin on cholesterol uptake, we used the more sensitive Flexstation3 assay and measured the intracellular BODIPY-cholesterol levels in response to different concentrations of laminarin. The results showed that laminarin significantly reduced cholesterol uptake in a concentration-dependent manner with an EC_50_ of 20.69 μM (Figure 2D). Ezetimibe, a known inhibitor of the NPC1L1 transporter, was used as a positive control in this assay.

### 2.2. Knockdown of NPC1L1 Attenuates the Inhibitory Effect of Laminarin on Cholesterol Uptake in Caco-2 Cells

The above results suggested that laminarin, a polysaccharide derived from brown seaweed, could be as effective as ezetimibe, a commonly used cholesterol-lowing drug. To explore the association between laminarin and the NPC1L1 transporter we used siRNA to knockdown NPC1L1 in Caco-2 cells and tested three siRNAs. The hNPC1L1-2732 siRNA was found to be the most efficient in reducing NPC1L1 expression (Figure 3A). We performed confocal imaging to visualize BODIPY-cholesterol uptake in native and NPC1L1-depleted Caco-2 cells treated with laminarin and ezetimibe individually at a concentration of 50 μM as well as in combination. Consistent with the data shown in Figure 2, our observations showed that in the native Caco-2 cells, both laminarin and ezetimib inhibited cholesterol uptake, and the combination of laminarin and ezetimibe (lam50 + eze50) demonstrated a competitive effect rather than a simple additive effect (Figure 3B,C). In contrast, when NPC1L1 expression was knocked down in Caco-2 cells, the tested compounds, laminarin and ezetimibe, failed to reduce cholesterol uptake. The same results were also confirmed using a more sensitive Flexstation3 assay (Figure 3D). These data support that laminarin reduces cholesterol uptake in Caco-2 cells by involving NPC1L1, which is similar to the mechanism of action of ezetimibe.

### 2.3. Laminarin Reduces Serum Lipid Concentration as Well as Subcutaneous and Visceral Fat Content in HFD-Fed Mice

We further examined the in vivo effects of laminarin on serum lipid and fat content in mice fed by a HFD for 14 weeks prior to treatments. The mice were divided into different groups, including low (50 mg/kg), medium (100 mg/kg), and high (200 mg/kg) doses of laminarin, 10 mg/kg ezetimibe, and the combination of medium-dose laminarin and ezetimibe (Eze10 + Lam100), and the treatments were administered daily intragastric gavage for 8 weeks. The food intake and body weight were regularly monitored through the experimental procedure (Appendix A), and the serum TC, TG, LDL-C, and HDL-C were compared between HFD and ND groups after 14 weeks to ensure the successful establishment of the hyperlipidemia model (Appendix A).

After 8 weeks of treatment with laminarin (50, 100, or 200 mg/kg) or ezetimibe (10 mg/kg), the serum TC in the ezetimibe group was reduced by 45.99% compared with that in the HFD-control group, and laminarin at all doses (low, medium, and high) and the combination of laminarin and ezetimibe (Eze10 + Lam100), also reduced serum TC with reductions of 15.72%, 26.14%, 44.56%, and 39.53%, respectively (Figure 4A). Serum TG levels were reduced by 33.04% in the ezetimibe group, as compared with the HFD-control group. In the high-dose laminarin and combined group (Eze10 + Lam100), serum TG was reduced by 27.04% and 34.31%, respectively (Figure 4B). However, there were no significant changes in the low- and medium-dose laminarin groups. The positive control ezetimibe reduced serum LDL-C by 44.72% compared to the HFD-control mice. Laminarin at medium and high doses reduced LDL-C by 27.12% and 40.31%, respectively, while the combination group showed a reduction of 63.57% (Figure 4C). There was no significant change in HDL-C levels in any group of the mice (Figure 4D).

Whole-body CT imaging analysis showed that HFD-control mice had significantly increased subcutaneous and visceral fat content compared with ND-control mice (Figure 4E). After daily intragastric administration of laminarin at a medium-dose, high-dose, and medium-dose in combination with ezetimibe for 8 weeks, the mice exhibited a reduction in subcutaneous and visceral fat content, but no significant changes in the low-dose laminarin group compared to HFD-control (Figure 4E).

### 2.4. Laminarin Reduces Hepatic Lipid Deposition in HFD-Fed Mice

To investigate the effect of laminarin on hepatic lipid accumulation, we evaluated the mouse liver morphology and histology. Liver tissues were dissected from mice after 8 weeks of laminarin treatment. The livers in the ND-control group appeared reddish-brown in color with a smooth surface by macroscopic observation. In contrast, the livers in the HFD-control group mice exhibited mild edema, softness and erosion, and a yellow-brown color, indicating the presence of fatty lesions (Figure 5A upper panel).

H&E staining of liver sections from ND-control mice showed no steatosis, no infiltration of inflammatory cells, and no other pathological changes in the hepatocytes. However, the livers of the HFD-control mice displayed edematous hepatocytes with diffuse large and small lipid droplets, indicative of steatosis. Treatment with laminarin at doses of 100 and 200 mg/kg/day visibly improved hepatic steatosis compared to the HFD-control group, showing minimal vesicular steatosis in hepatocytes. In combination with ezetimibe, laminarin at 100 mg/kg/day further ameliorated the pathological changes induced by a HFD, with significant improvement in hepatic cell steatosis (Figure 5A, lower panel).

The levels of TC, TG, and TBA in the livers of the HFD-control group were 1.7, 1.7, and 2.0 times higher, respectively, than those in the ND-control group (Figure 5B–D). After 8 weeks of treatment with intragastric administration, the hepatic TC of mice in the positive control ezetimibe group was reduced by 28.85% compared with that in the HFD-control group. High-dose laminarin (200 mg/kg) and the combination medium-dose laminarin (100 mg/kg) with ezetimibe led to reductions in hepatic TC by 20.80% versus 19.82%, respectively. However, no significant changes were observed in the other treatment groups (Figure 5B). Hepatic TG were reduced by 32.03% in the ezetimibe group compared with the HFD-control group. Treatment with 200 mg/kg of laminarin and in combination with ezetimibe resulted in a reduction in hepatic TG by 26.31% versus 40.25%, respectively; however, no significant changes in hepatic TG levels in the other treatment groups (Figure 5C).

Hepatic total bile acids (TBA) were reduced by 26.22% in the positive control ezetimibe compared to the HFD-control. The high-dose laminarin and medium-dose laminarin combined with ezetimibe showed reductions in hepatic TBA by 21.79% and 27%, respectively. No significant changes in hepatic TBA were observed in the other treatments (Figure 5D). Collectively, these results obtained from H&E staining and determination of TC, TG, and TBA levels in mouse liver indicate that laminarin effectively ameliorate aberrant lipid parameters induced by a HFD in mice and also reduces hepatic TC, TG, and TBA levels.

### 2.5. Laminarin Increases Fecal Cholesterol in HFD-Fed Mice

During the experiment, fecal TC, TG, and TBA excretion in HFD-control mice increased by 165%, 532%, and 58%, respectively, compared with those in control-ND mice (Figure 6A–C). After 8 weeks of treatment with laminarin or ezetimibe via intragastric administration, fecal TC excretion increased by 34.91% in the ezetimibe group (positive control) compared with the HFD-control group. The high-dose laminarin (200 mg/kg) and the combination of medium-dose laminarin (100 mg/kg) with ezetimibe (10 mg/kg) increased fecal TC excretion by 37.7% and 34.1%, respectively (Figure 6A). No significant change was observed in fecal TG and TBA excretion in the laminarin-treated mice compared to the HFD-control group (Figure 6B,C). These results suggest that 200 mg/kg laminarin and the combination of 100 mg/kg laminarin and ezetimibe further increased HFD-induced fecal cholesterol excretion to a similar extent.

### 2.6. Laminarin Ameliorates Jejunal Villus Breakage and Downregulates NPC1L1 Expression in the Jejunum of HFD-Fed Mice

We also examined the in vivo expression of the NPC1L1 protein, primarily localized in the jejunum. First, the histology of the mouse jejunum was evaluated. Jejunal tissues were sectioned after 8 weeks of treatment, and H&E staining results showed that the villi in the jejunum of control mice were well-arranged and structurally intact, with no damage to the mucosa. In the jejunum of HFD-control mice, some villi appeared swollen and necrotic, shortened, and loosened, and the normal mucosa layer structure was absent at the necrotic site. Laminarin administration significantly improved jejunal villus breakage and intestinal structural damage (Figure 7A).

Subsequently, we examined the NPC1L1 expression in the jejunum. The results showed that NPC1L1 expression in the jejunum of HFD-control mice was upregulated 1.08-fold compared with the ND-control group. In comparison to the HFD-control group, NPC1L1 was downregulated by 41.5% in the jejunum of mice treated with ezetimibe. In the jejunum of mice treated with 200 mg/kg or 100 mg/kg of laminarin in combination with ezetimibe, NPC1L1 was downregulated by 30.3% and 46.4%, respectively, with no significant changes in other treatment groups (Figure 7B). These results demonstrate that laminarin ameliorates jejunal villus breakage and downregulates NPC1L1 expression in the jejunum of HFD-fed mice.

## 3. Discussion

This study aimed to investigate the lipid-lowering potential of laminarin, a natural bioactive molecule derived from edible brown algae. Our findings show that laminarin can counteract diet-induced hyperlipidemia by reducing intestinal cholesterol uptake, possibly through the down-regulation of NPC1L1 expression, a key transporter for cholesterol uptake.

Previous studies have shown that another polysaccharide from brown algae, fucoidan, inhibits atherosclerosis by regulating the expression of genes involved in cholesterol reverse transport (RCT), such as ABCA1, SR-A1, and PPAR [24]. Fucoidan has also been found to reduce NPC1L1 expression, inhibit cholesterol uptake in the intestine, and prevent hyperlipidemia-induced atherosclerosis in apoE^−/−^ mice [25]. To explore the effects of laminarin, also derived from brown algae, on RCT, we monitored cholesterol efflux in J774A.1 murine macrophage. However, laminarin did not show a significant effect on cholesterol efflux (Appendix A). Nevertheless, when using Caco-2 cells to simulate cholesterol transport in the small intestinal epithelium, we found that laminarin effectively inhibited cholesterol uptake in a concentration-dependent manner. The results also showed that the effect of laminarin at 50 μM has reached a plateau, and even if the concentration continues to increase, the effect could not be further improved. That might be because there was a plateau in the dose-effect relationship of laminarin. When the compound reaches a certain threshold of concentration, further increasing its concentration will not improve its efficacy and may even produce toxic effects. Additionally, 50 μM laminarin and 50 μM ezetimibe exhibited competitive effects on cholesterol uptake in Caco-2 cells. These results led us to speculate that laminarin may act by inhibiting NPC1L1 in the small intestine.

We then evaluated the lipid-lowering effect and mechanism of laminarin in vivo using HFD mice. Laminarin effectively reduced serum TC, TG, and LDL-C levels in mice. In this study, laminarin at 100 mg/kg with ezetimibe at 10 mg/kg rather than laminarin at 200 mg/kg with ezetimibe at 10 mg/kg was used to ascertain the full therapeutic potential of these compounds. One would expect that the combination of 200 mg/kg laminarin with 10 mg/kg ezetimibe may produce a stronger effect than 100 mg/kg laminarin with ezetimibe at 10 mg/kg. However, this only occurs when the binding sites of NPC1L1 are not fully saturated. Since laminarin and ezetimibe were speculated to act on the same target, they compete with each other for binding sites on NPC1L1. Therefore, there is a competitive inhibitory effect between them rather than a synergistic or additive effect. Unlike fucoidan [26], laminarin did not affect HDL-C levels. This finding aligns with our hypothesis that laminarin shares mechanistic similarities with ezetimibe, as ezetimibe has also been shown to reduce intestinal cholesterol uptake without affecting serum HDL-C levels [27]. The competitive inhibitory effect of laminarin and ezetimibe on TC and TG levels in vivo mirrored the results obtained in the in vitro experiments, except for LDL-C, where an additive inhibitory effect was observed. Polysaccharides have the ability to create a water layer barrier in the intestine and facilitate the excretion of bile acids. This process promotes the conversion of hepatic cholesterol into bile acids at an increased rate, thereby reducing the cholesterol content in the liver and lowering serum LDL-C concentration [28,29]. These findings suggest that the mechanisms of action of laminarin and ezetimibe are not identical, and that multiple effects may contribute to the observed in vivo results.

It is widely recognized that increased excretion in fecal lipids is associated with lower levels of serum lipids, which helps mitigate the accumulation of fat in the liver [30]. Our results demonstrated that laminarin significantly increases fecal excretion of TC in mice, while not significantly affecting fecal levels of TG or TBA levels. In the mouse liver, laminarin effectively reduced the levels of TC, TG, and TBA and improved the diffuse vesicular steatosis of hepatocytes in HFD-mice. We also observed that laminarin significantly ameliorated jejunal villus breakage and intestinal structural damage. It also helped reduce the accumulation of subcutaneous and visceral fat in HFD-mice, leading to partial restoration of pathological changes. Hyperlipidemia has been associated with increased levels of TNF-α, which promotes a chronic inflammatory state and can impair intestinal barrier function [31]. Therefore, it would be interesting to explore the potential effect of laminarin on intestinal inflammation in future research.

NPC1L1 is the most abundant protein in the epithelium of the proximal segment of the jejunum, which explains the active uptake of cholesterol by this part of the intestine, particularly the proximal jejunum, compared with other regions of the gut [18,32]. The lipid-lowering effect of ezetimibe is largely attributed to its inhibition of NPC1L1 transporter activity. In addition, ezetimibe has been shown to downregulate NPC1L1 protein expression [20,33], although this downregulation is not as direct and efficient as inhibiting the transporter activity itself. Similar to ezetimibe, our study showed that laminarin can downregulate NPC1L1 protein levels. This effect on the NPC1L1 transporter allows laminarin to rapidly and effectively reduce serum lipid levels induced by a HFD in mice. Considering the significant improvements observed in blood lipids, fecal lipids, and hepatic lipids due to laminarin treatment but the lack of inhibition in cholesterol efflux from macrophages (Appendix A), which is an initial step in RCT, it is plausible to speculate that laminarin may also interact with other regulatory players involved in RCT, such as ABCG5/G8, to maintain cholesterol homeostasis. However, further experimental evidence is required to confirm this speculation.

## 4. Materials and Methods

### 4.1. Chemicals and Reagents

A small interfering RNA oligonucleotide duplex targeting NPC1L1 was designed and synthesized by Sangon Biotech Co., Ltd. (Shanghai, China). Laminarin (purity: ≥96%) and ezetimibe (purity: ≥99%) were acquired from Macklin Biochemical Technology Co., Ltd. (Shanghai, China). Fetal bovine serum (FBS) was supplied by PAN-SERATECH (Adenbach, Germany). Minimum essential medium (MEM), sodium pyruvate, MEM non-essential amino acids solution (NEAA), benzylpenicillin–streptomycin, lipofectamine 3000, minus serum medium (Opti-MEM), and PBS were obtained from Gibco (Waltham, MA, USA). Resazurin sodium salt, dimethyl sulfoxide (DMSO), and sodium carboxymethyl cellulose (CMC-Na) were purchased from Sigma-Aldrich (St. Louis, MO, USA), and digitonin was obtained from TOPSCIENCE Biotechnology Co., Ltd. (Shanghai, China). Modified Harris hematoxylin solution, eosin Y solution, poly-D-lysine and 4′,6-diamidino-2-phenylindole (DAPI) were bought from Solarbio Science & Technology Co., Ltd. (Beijing, China).

The normal diet and the customized high-fat diet were purchased from Keao Xieli Feed Co., Ltd. (Beijing, China). RIPA lysis and extraction buffer, halt protease and phosphatase inhibitor cocktail, protein ladder, NuPAGE LDS sample buffer, SuperSignal west pico plus chemiluminescent substrate, BCA protein assay kit, and histoplast paraffin were obtained from Thermo Scientific (Waltham, MA, USA). Polyvinylidene fluoride (PVDF) membrane was provided by GE Healthcare Life Sciences (Pittsburgh, PA, USA). Nonfat dry milk (NFDM) was purchased from Cell Signaling Technology (Beverly, MA, USA). The primary antibody against NPC1L1 was obtained from Novus Biologicals (Littleton, CO, USA). The anti-GAPDH antibody and goat anti-rabbit IgG H&L (HRP) were products of Abcam (Cambridge, UK). Assay kits for measuring total cholesterol (T-CHO), triglyceride (TG), low-density lipoprotein cholesterol (LDL-C), high-density lipoprotein cholesterol (HDL-C), and total bile acid (TBA) were purchased from JianCheng Bioengineering Institute (Nanjing, China).

The synthesis of BODIPY-cholesterol involved the use of the following reagents: 3β-hydroxy-Δ5-cholenoic acid and 2,4-dimethylpyrrole were acquired from Macklin Biochemical Technology Co., Ltd. (Shanghai, China). 4-dimethylaminopyridine was purchased from Bide Pharmaceutical Technology Co., Ltd. (Shanghai, China). Boron trifluoride diethyl ether, pyridine, oxalyl chloride, and anhydrous Na_2_SO_4_ were acquired from Aladdin Bio-Chem Technology Co., Ltd. (Shanghai, China). Muriatic acid, ethyl acetate, dichloromethane, methyl alcohol, petroleum ether, and acetic anhydride were purchased from China National Pharmaceutical Group Co., Ltd. (Beijing, China). Triethylamine and dimethyl sulfoxide-d6 were purchased from Titan Scientific Co., Ltd. (Shanghai, China).

### 4.2. Fluorescent BODIPY-Cholesterol Synthesis

Acetic anhydride (1.66 mmol), 3β-hydroxy-Δ5-cholenoic acid (1.33 mmol), and 4-dimethylaminopyridine (0.133 mmol) were dissolved in pyridine (10 mL) and stirred for 4 h at room temperature. After the reaction, iced water (10 mL) was added to quench the reaction. Then, 3 N HCl (3 mmol) was added dropwise with stirring to adjust the pH of the solution from 5 to 2. The mixture was extracted with ethyl acetate (3 × 20 mL), and the organic phases were combined before being washed with saturated aqueous NaCl (2 × 20 mL) and filtered and dried over anhydrous Na_2_SO_4_. After concentration, compound **1** was isolated as white solid powder by column chromatography using CH_2_Cl_2_/MeOH (60:1) as the eluent. Compound **1** (0.82 mmol) was dissolved in CH_2_Cl_2_ (30 mL) in an ice bath under nitrogen, and oxalyl chloride (3.28 mmol) was added. The mixture was stirred overnight at room temperature, and compound **2** was obtained as a white solid after concentration. Compound **2** was dissolved in CH_2_Cl_2_ (10 mL), and 2,4-dimethylpyrrole (1.804 mmol) was added. The reaction mixture was heated under reflux under nitrogen for 4 h. After cooling to room temperature, boron trifluoride diethyl ether (4.92 mmol) and triethylamine (3.28 mmol) were added, and the mixture was stirred under nitrogen overnight. After completion of the reaction, 5 mL of water was added, and then the mixture was extracted before the organic phase was collected and dried. The filtrate was concentrated and purified by column chromatography using EtOAc/petroleum ether (1:10) as the eluent. A total of three synthesis cycles resulted in a yield of 608.8 mg of compound **3** (BODIPY-cholesterol) with fluorescence activity (Appendix A), yielding approximately 65% to 79% (Figure 1).

3β-Acetoxy-23-(4,4-difluoro-1,3,5,7-tetramethyl-4-bora-3a,4a-diaza-sindacene-8-yl)-24-norchol-5-ene (compound **3**, BODIPY-cholesterol): ^1^H NMR (500 MHz, DMSO-d6) δ 6.06 (s, 2H), 5.38 (d, *J* = 5.0 Hz, 1H), 4.62 (dd, *J* = 11.4, 6.1 Hz, 1H), 3.16 (td, *J* = 12.9, 5.5 Hz, 1H), 2.72 (td, *J* = 12.8, 3.1 Hz, 1H), 2.53 (s, 5H), 2.44 (s, 5H), 2.34 (s, 1H), and 2.06 (s, 1H).

### 4.3. Cell Culture and Resazurin Conversion Assay

Caco-2 cells were obtained from the Cell Bank of the Chinese Academy of Sciences and cultured in 25 cm^2^ cell culture flasks with minimum essential medium supplemented with 1 mM NEAA, 1 mM sodium pyruvate, 100 U/mL benzylpenicillin, 100 µg/mL streptomycin, and 20% FBS. Cells were maintained at 37 °C with 5% CO_2_ in a humidified atmosphere. For the experiments, Caco-2 cells were seeded at a density of 0.2 × 10^6^ per mL in 6-well-plates or 96-well-plates for experiments.

Caco-2 cells were treated with digitonin (50 µg/mL) and laminarin (3, 10, 30, 50, 70, and 100 µM) for 24 h. After incubation, cells were washed with PBS and incubated with resazurin for another 4 h. Fluorescence emitted by the generated resorufin was quantified using excitation/emission wavelengths of 535/580 nm as a measurement of cell viability.

### 4.4. Cholesterol Uptake Assay

First, Caco-2 cells in the logarithmic growth phase were harvested and seeded at a density of 0.2 × 10^6^ cells/mL in a 96-well-plate in a total volume of 100 μL medium/well using a Vi-Cell XR cell counter (Beckman Coulter, Indianapolis, IN, USA). After 24 h incubation, Caco-2 cells were treated with ezetimibe (50 µM), various concentrations of laminarin (5, 10, 25, 50, 100, and 150 µM) and a solvent vehicle control (DMSO) for 24 h. Then, the medium was discarded and washed once with PBS to remove non-adherent or dead cells before being replaced with the MEM medium containing 0.2% BSA. Caco-2 cells were labeled with BODIPY-cholesterol (0.0625 mM) and treated with ezetimibe (50 µM), various concentrations of laminarin (5, 10, 25, 50, 100, and 150 µM) and a solvent vehicle control (DMSO) for another 24 h. After the incubation, the extracellular fluid was discarded, and 100 µL of 10% SDS were added to each well before being shaken for 6 min, and 80 µL of cell lysates were collected into a black 96-well plate. The fluorescence value (Ex/Em = 482/515 nm) was detected using the Flexstation3 (Molecular Devices, San Jose, CA, USA), and the fluorescence values were normalized to the cell number in each well.

### 4.5. Rapid High-Resolution Confocal Microscope Imaging

Caco-2 cells were seeded at a density of 1 × 10^5^ cells/mL in a 35 mm laser confocal culture dish coated with 0.05 mg/mL poly-D-lysine. After incubation for 24 h, Caco-2 cells were treated with ezetimibe (50 µM), various concentrations of laminarin (5, 10, 25, 50, 100, and 150 µM) and a solvent vehicle control (DMSO) for 24 h. Then, the medium was discarded and washed once with PBS to remove non-adherent cells or dead cells before replacement of the MEM medium containing 0.2% BSA. Caco-2 cells were labeled with BODIPY-cholesterol (0.0625 mM) and treated with ezetimibe (50 µM), various concentrations of laminarin (5, 10, 25, 50, 100, and 150 µM) and a solvent vehicle control (DMSO) for another 24 h. Then the medium was discarded, and cells were fixed with paraformaldehyde cell fixative for 20 min. After fixation, 10 µg/mL of DAPI was added to cells for nuclear staining. After 10 min, the supernatant was discarded, and cells were washed with PBS and imaged with a confocal microscope (Leica STELLARIS 5, Leica, Wetzlar, Germany).

### 4.6. Design and Synthesis of siRNA

A small interfering RNA oligonucleotide duplex targeting NPC1L1 Table 1 was designed and synthesized by Sangon Biotech Co., Ltd. (Shanghai, China).

**Table 1 marinedrugs-21-00624-t001:** Genetic sequence of NPC1L1 siRNA oligonucleotide duplex.

Gene ID	Genetic Sequence	
	Sense (5′-3′)	Antisense (5′-3′)
hNPC1L1-176	CUAUGACGAAUGUGGGAAGAATT	UUCUUCCCACAUUCGUCAUAGTT
hNPC1L1-2732	CCUCUUUCUGAACCGCUACUUTT	AAGUAGCGGUUCAGAAAGAGGTT
hNPC1L1-1980	GCCACCAGCUACAUUGUCAUATT	UAUGACAAUGUAGCUGGUGGCTT
NC	UUC UCC GAA CGU GUC ACG UTT	ACG UGA CAC GUU CGG AGA ATT
FAM-NC	UUC UCC GAA CGU GUC ACG UTT	ACG UGA CAC GUU CGG AGA ATT

### 4.7. Transfection of Caco-2 Cells

Caco-2 cells (0.2 × 10^6^ cells/well) were plated on 6 well-plates and incubated for 24 h. Cells were transfected with small interfering RNA (siRNA) against NPC1L1 (200 nM) using Lipofectamine 3000 and minus serum medium (Opti-MEM) according to the manufacturer’s recommendations. Cells were harvested 48 h after transfection for further experiments.

### 4.8. Animal Experiments

Eighty-four male C57BL/6J mice (pathogen-free; 6 weeks of age) were purchased from Vital River Laboratory Animal Technology Co., Ltd. (Beijing, China). Throughout the study, the animals were housed in plastic mouse cages at a controlled temperature of 23 ± 2 °C with 12/12 h light-dark cycle and 50 ± 10% humidity. Mice had free access to water and food. The in vivo study was conducted in accordance with the Animal Center of Medical College of Qingdao University according to the guidelines and regulations of the Laboratory Animal Care and Use Committee of Qingdao University (protocol code: No.20220504C579120221220051, 21 April 2022). The mice were acclimated for 1 week to stabilize their metabolic conditions.

Mice were monitored for body weight and food intake. Mice were fed either a normal diet (ND) or a high-fat diet (HFD). HFD consists of 10% lard, 0.5% bile salt (pig), and 3% cholesterol mixed with ND, as formerly described [34]. After feeding the mice with ND or HFD for 14 weeks, blood samples were collected from the tail vein, and blood lipid biochemical indexes were determined to confirm the establishment of a hyperlipidemia model. Then, the mice were randomly divided into seven groups, 12 for each group, and the treatment dose of compounds was determined according to previous studies [16,35]. The seven treatment groups were as follows: ND, control group; HFD, control group; HFD-ezetimibe group (10 mg/kg/day); and HFD high-dose laminarin (200 mg/kg/day); HFD medium-dose laminarin (100 mg/kg/day); HFD low-dose laminarin (50 mg/kg/day); and HFD drug combination group (Ezetimibe: 10 mg/kg/day and laminarin: 100 mg/kg/day). The mice in the vehicle control group were given the same volume of 0.3% CMC-Na by gavage, and all animals in all groups were gavaged once a day for 8 weeks with their respective treatments. During the treatment period, food intake was recorded daily, and the mouse weight was measured weekly. At the end of the experiment, fresh feces were collected and immediately stored at −80 °C for subsequent analysis. Mice were fasted for 12 h, blood samples were collected, centrifuged at 1342× *g* for 15 min at 4 °C, and serum was stored at −80 °C for further analysis. After the mice were euthanized by isoflurane anesthesia, samples such as liver and jejunum were dissected for subsequent analysis.

### 4.9. Analysis of Serum, Hepatic and Fecal Lipids

After a 12 h fasting period, mice were anesthetized, and blood samples were collected through eye bleeding, and kept motionless for 30 min. The whole blood was centrifuged at 1342× *g* for 15 min to separate the serum. The serum levels of TG, TC, TBA, HDL-C, and LDL-C were determined in a microplate reader (Tecan SUNRISE, Grödig, Austria) using corresponding commercial assay kits. Liver and feces samples were accurately weighed and mechanically homogenized under ice-water bath conditions. Then, the homogenized samples were centrifuged at 685× *g* for 10 min, and the resulting supernatant was used to measure the same biochemical markers mentioned above using the corresponding assay kits.

### 4.10. Hematoxylin and Eosin (H&E) Staining

After the mice were sacrificed, a 0.5 cm section of jejunal tissue and a lobular section of the liver were fixed in 4% paraformaldehyde for at least 24 h. For dehydration of specimens, graded alcohols (70~100%, 5% a grade) were applied before clearing and infiltration in xylene. Tissue sections were embedded in paraffin wax and cut into 6 μm thick using a rotary microtome (Leica RM2255, Leica, Wetzlar, Germany). Tissue sections were attached to poly-L-lysine-treated glass slides. After deparaffinization in xylene and rehydration in graded alcohols (100~70%), tissue samples were stained with modified Harris hematoxylin solution for 6 min and counterstained with Eosin Y solution for 3 min. Excessive eosin of tissue sections was removed with alcohol and subsequent clearing in xylene. Finally, images of the tissue sections were captured using a microscope (Nikon Eclipse TS100, Nikon, Tokyo, Japan).

### 4.11. Protein Extraction and Western Blotting

Jejunal tissue samples were homogenized in RIPA lysis and extraction buffer containing 1% protease and phosphatase inhibitor cocktail (Thermo Scientific, Waltham, MA, USA) using a high flux tissue grinder (SCIENTZ-48, SCIENTZ, Zhejiang, China). After full lysis for 30 min at 4 °C using a rotary mixer (MIULAB, MIULAB, Hangzhou China), the tissue lysate was centrifuged at 15,777× *g* for 15 min to collect the supernatant. Total protein concentrations in the supernatant were detected using a BCA protein analysis kit. The protein samples were separated in 10% SDS-PAGE gels in tris-glycine buffer and then transferred to PVDF membranes. The PVDF membrane was blocked in TBS-T (24.9 mM Tris, 137 mM NaCl, 2.7 mM KCl, 0.1% Tween-20, pH 7.4) with 5% nonfat dry milk for 1 h at room temperature, followed by incubated with primary antibody at 4 °C overnight. The primary antibodies against NPC1L1 (#NB400-127), and GAPDH (#ab181602) were used. Membranes were incubated with a goat anti-rabbit IgG H&L (HRP) at room temperature for 1 h. Protein bands were visualized by using a chemiluminescent substrate kit. Finally, Image Lab software (Bio-Rad Laboratories, version 5.2.1, Hercules, CA, USA) was used to quantify the relative protein expression levels. The target protein abundance was normalized to the reference protein GAPDH, and then the values of all groups were normalized to the ND group.

### 4.12. Living Small Animal CT Imaging

After eight weeks of laminarin treatment, a subgroup of three mice from each group was randomly selected and anesthetized with inhaled isoflurane gas using a small animal anesthesia machine (RWD R500, Shenzhen, China). After the animals were completely anesthetized, a whole-body scan was conducted using a Living small animal CT instrument (PerkinElmer Quantum GX2, Waltham, MA, USA), and images of the subcutaneous and visceral fat were captured.

### 4.13. Statistical Analysis

All data were presented as the mean ± SD, and statistical analysis was performed using GraphPad Prism 7.00 software. Student’s *t*-tests and one-way ANOVA corrected by the Bonferroni test were used to assess statistical differences between groups, and a value of *p* < 0.05 was considered statistically significant.

## 5. Conclusions

This study provides evidence that laminarin, with MFH properties, has the potential for the prevention and treatment of hypercholesterolemia and related cardiovascular diseases. The mechanisms involved in this potential therapeutic effect include the reduced expression of NPC1L1, a key transporter involved in cholesterol uptake. By downregulating NPC1L1, laminarin contributes to the inhibition of intestinal cholesterol absorption, and, subsequently, lowers serum lipid levels. These findings highlight the potential of laminarin as a natural compound for managing hypercholesterolemia and its associated cardiovascular risks. Further research is needed to evaluate the clinical implications of laminarin in the context of hypercholesterolemia.

## Data Availability

The original data are available on request.

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
