# Peer review of "Laminarin Reduces Cholesterol Uptake and NPC1L1 Protein Expression in High-Fat Diet (HFD)-Fed Mice"

_marinedrugs, 2023, doi:10.3390/md21120624_

Round 1

Reviewer 1 Report

Comments and Suggestions for Authors

Thank you for providing opportunity to review this article.

The manuscript titled, ‘Laminarin Reduces Cholesterol Uptake and NPC1L1 Protein 2 Expression in High-fat Diet (HFD)-fed Mice’ by He Z., et al have shown the effects of laminarin on cholesterol uptake by using invitro and in vivo experiments. Authors show the cholesterol uptake reduces in caco2 cells by observing reduced fluorescence intensity of cholesterol in laminarin dose dependent manner. In vivo data suggest reduced serum TC, TG and hepatic bile acids with treatment of laminarin in HFD diet fed mice. Serum lipid, subcutaneous and visceral fat content in HFD is reduced in HFD mice after laminarin treatment. It also shows reduction in hepatic lipid deposition in dose dependent manner. Figure 6 show laminarin increases fecal lipid deposition in HFD fed mice. With H & E staining authors show laminarin ameliorates the jejunal villus breakage and western data suggests NPC1L1 downregulation in the jejunum of HFD-fed mice.

The results are clear and presented well. However, authors need solid evidence to claim that laminarin can downregulate NPC1L1 expression. Relative m-RNA levels before and after the laminarin treatment could be checked.

The article is well structured into sections and within the scope of the journal. Apart for the above comments the manuscript is clear, comprehensive and relevant to the dyslipidemia treatment field. The flow of the manuscript is smooth. Important articles in the recent past have been included and justified in the manuscript. There are no excessive self-citations. The data provided in the current manuscript is of great importance, relevance and a valuable addition to the existing knowledge of the scientific community.

Author Response

Comments and Suggestions for Authors

Thank you for providing opportunity to review this article. The manuscript titled, ‘Laminarin Reduces Cholesterol Uptake and NPC1L1 Protein 2 Expression in High-fat Diet (HFD)-fed Mice’ by He Z., et al have shown the effects of laminarin on cholesterol uptake by using invitro and in vivo experiments. Authors show the cholesterol uptake reduces in caco2 cells by observing reduced fluorescence intensity of cholesterol in laminarin dose dependent manner. In vivo data suggest reduced serum TC, TG and hepatic bile acids with treatment of laminarin in HFD diet fed mice. Serum lipid, subcutaneous and visceral fat content in HFD is reduced in HFD mice after laminarin treatment. It also shows reduction in hepatic lipid deposition in dose dependent manner. Figure 6 shows laminarin increases fecal lipid deposition in HFD fed mice. With H & E staining authors show laminarin ameliorates the jejunal villus breakage and western data suggests NPC1L1 downregulation in the jejunum of HFD-fed mice.

The results are clear and presented well. However, authors need solid evidence to claim that laminarin can downregulate NPC1L1 expression. Relative m-RNA levels before and after the laminarin treatment could be checked.

The article is well structured into sections and within the scope of the journal. Apart for the above comments, the manuscript is clear, comprehensive and relevant to the dyslipidemia treatment field. The flow of the manuscript is smooth. Important articles in the recent past have been included and justified in the manuscript. There are no excessive self-citations. The data provided in the current manuscript is of great importance, relevance and a valuable addition to the existing knowledge of the scientific community.

Response: Thank you very much for your kind evaluation of our work. We agree with your suggestion that solid evidence should be provided to confirm that laminarin can downregulate NPC1L1 expression. However, instead of testing the mRNA levels of NPC1L1, we performed western-blotting analysis to evaluate its protein levels. As shown in the figure below, NPC1L1 protein levels had been significantly downregulated after laminarin treatment.

Figure. Inhibition of NPC1L1 protein expression by laminarin in Caco-2 cells (n=5). All data were presented as the mean ± SD, *p < 0.05, **p < 0.01, n.s. no significance (one-way ANOVA).

Reviewer 2 Report

Comments and Suggestions for Authors

This is a very interesting and well performed study showing the effect of laminarin, a polysaccharide found in Brown Algae, on cholesterol absorption and disposition. The study was performed by using both in vitro and in vivo experiments. The results are convincing and demonstrate an inhibitory effect of laminarin on NPC1L1, target of ezetimibe, which reduced cholesterol uptake in Caco-2 cells and in vivo in C57BL/6 mice fed high-fat diet.

Minor comment:

please rephrase the following sentence: Nevertheless, when using Caco 2 cells simulating the process of cholesterol uptake and transport across the small intestinal epithelium, we found that laminarin effectively inhibited cholesterol uptake in a concentration-dependent manner.

Author Response

Comments and Suggestions for Authors

This is a very interesting and well performed study showing the effect of laminarin, a polysaccharide found in Brown Algae, on cholesterol absorption and disposition. The study was performed by using both in vitro and in vivo experiments. The results are convincing and demonstrate an inhibitory effect of laminarin on NPC1L1, target of ezetimibe, which reduced cholesterol uptake in Caco-2 cells and in vivo in C57BL/6 mice fed high-fat diet.

Minor comment:

please rephrase the following sentence: Nevertheless, when using Caco 2 cells simulating the process of cholesterol uptake and transport across the small intestinal epithelium, we found that laminarin effectively inhibited cholesterol uptake in a concentration-dependent manner.

Response: Thank you for your recognition of our work. The sentence has been rephrased as: Nevertheless, when using Caco-2 cells to simulate cholesterol transport in the small intestinal epithelium, we found that laminarin effectively inhibited cholesterol uptake in a concentration-dependent manner.

Reviewer 3 Report

Comments and Suggestions for Authors

I would like to offer some points of clarification and direct a few inquiries to the authors:

1. There is a notable lack of consistency in the abbreviation and expansion of terms throughout the manuscript, notably, such as in lines 18 (NPC1L1), and 23 (TC, TG, LDL-C). This must be rectified in accordance with academic writing conventions.

2. The segmentation of the introduction into disparate paragraphs addressing cardiovascular diseases, laminarin, and NPC1L1 lacks synthesis and scholarly coherence. A comprehensive rewrite is required to integrate these concepts effectively.

3. The absence of a dose-response effect of laminarin in Figure 1C, is concerning. An explanation grounded in experimental data or theoretical analysis is necessary.

4. The justification for administering laminarin to mice without dose calculation from cell-based experimental results, instead of deferring to dosages from prior studies, is questionable. This approach undermines the study's experimental design.

5. The selection of dosages for laminarin and ezetimibe in this study warrants further justification. The singular use of laminarin at 100 mg/kg combined with ezetimibe at 10 mg/kg raises questions when the efficacy of laminarin at 200 mg/kg alone appears superior. One would expect a synergistic or additive effect with the combination treatment, yet this is not reflected in your results. It is imperative to explore and present findings for the combination of laminarin at 200 mg/kg with ezetimibe at 10 mg/kg to ascertain the full therapeutic potential of these compounds. The omission of this data is a notable gap in the research that must be addressed.

6. In Figure 4C, the combination of laminarin 100 mg/kg with ezetimibe 10 mg/kg underperforms compared to ezetimibe 10 mg/kg monotherapy in reducing TC levels. This counterintuitive result requires a robust scientific explanation.

The manuscript, in its current form, does not meet the high standards of this journal. Addressing these points with due diligence is crucial for further consideration.

Comments on the Quality of English Language

Moderate editing of the English language is required.

Author Response

Question #1: There is a notable lack of consistency in the abbreviation and expansion of terms throughout the manuscript, notably, such as in lines 18 (NPC1L1), and 23 (TC, TG, LDL-C). This must be rectified in accordance with academic writing conventions.

Response #1: Thanks for your suggestion. We have rectified and labeled the inconsistent abbreviations and expansions throughout the manuscript in this revision.

Question #2: The segmentation of the introduction into disparate paragraphs addressing cardiovascular diseases, laminarin, and NPC1L1 lacks synthesis and scholarly coherence. A comprehensive rewrite is required to integrate these concepts effectively.

Response #2: Yes, we have added a comprehensive description of them in this revision.

Question #3: The absence of a dose-response effect of laminarin in Figure 1C, is concerning. An explanation grounded in experimental data or theoretical analysis is necessary.

Response #3: Thanks for your suggestion. Actually, we found that, within a certain concentration range, the effect of laminarin enhances with the increase of laminarin, and reduces with the decrease of laminarin. It was also found that there was a plateau in the dose-effect relationship of laminarin. When the compound reaches a certain threshold of concentration, further increasing its concentration will not improve its efficacy, and may even produce toxic effects. Therefore, our results show that the effect of laminarin at 50 μM has reached a plateau, and even if the concentration continues to increase, the effect could not be further improved.  

Question #4:  The justification for administering laminarin to mice without dose calculation from cell-based experimental results, instead of deferring to dosages from prior studies, is questionable. This approach undermines the study's experimental design.

Response #4: In practice, to determine the dose of laminarin used in in vivo experiments, we referred to this paper: Zha X Q, Zhang W N, Peng F H, et al. Alleviating VLDL overproduction is an important mechanism for Laminaria japonica polysaccharide to inhibit atherosclerosis in LDLr(-/-) mice with diet-induced insulin resistance[J]. Mol Nutr Food Res, 2017, 61(4). For the positive control, ezetimibe, we referred to this paper to ascertain the dose for in vivo experiments: Xie P, Jia L, Ma Y, et al. Ezetimibe inhibits hepatic Niemann-Pick C1-Like 1 to facilitate macrophage reverse cholesterol transport in mice[J]. Arterioscler Thromb Vasc Biol, 2013, 33(5): 920-925.

Question #5: The selection of dosages for laminarin and ezetimibe in this study warrants further justification. The singular use of laminarin at 100 mg/kg combined with ezetimibe at 10 mg/kg raises questions when the efficacy of laminarin at 200 mg/kg alone appears superior. One would expect a synergistic or additive effect with the combination treatment, yet this is not reflected in your results. It is imperative to explore and present findings for the combination of laminarin at 200 mg/kg with ezetimibe at 10 mg/kg to ascertain the full therapeutic potential of these compounds. The omission of this data is a notable gap in the research that must be addressed.

Response #5: We agree with your suggestion that the combination of 200 mg/kg laminarin and 10 mg/kg ezetimibe may produce stronger effect than that of 100 mg/kg laminarin and ezetimibe. However, this only occurs when the binding sites of NPC1L1 is not fully saturated. Because laminarin and ezetimibe act on the same target to reduce blood lipids. The two compounds compete with each other for binding sites on NPC1L1. Therefore, there is a competitive inhibitory effect between them, rather than a synergistic or additive effect.

Question #6: In Figure 4C, the combination of laminarin 100 mg/kg with ezetimibe 10 mg/kg underperforms compared to ezetimibe 10 mg/kg monotherapy in reducing TC levels. This counterintuitive result requires a robust scientific explanation.

Response #6: The results in Figure 4C can be explained that laminarin and ezetimibe have competitive inhibition when applied in combination. This illustrates that these two compounds act on the same target of NPC1L1.

Round 2

Reviewer 3 Report

Comments and Suggestions for Authors

Please include explanations for Question 3 and Question 5 in the final accepted version.

Author Response

Comments and Suggestions: Please include explanations for Question 3 and Question 5 in the final accepted version.

Response: Thanks for your suggestion. We have included and labeled the explanations for Question 3 and Question 5 in this revision.